# Robust PCA with compressed data

**Wooseok Ha**
University of Chicago
haywse@uchicago.edu

**Rina Foygel Barber**
University of Chicago
rina@uchicago.edu

## Abstract

The robust principal component analysis (RPCA) problem seeks to separate low-rank trends from sparse outliers within a data matrix, that is, to approximate a $n \times d$ matrix $D$ as the sum of a low-rank matrix $L$ and a sparse matrix $S$. We examine the robust principal component analysis (RPCA) problem under data compression, where the data $Y$ is approximately given by $(L+S) \cdot C$, that is, a low-rank + sparse data matrix that has been compressed to size $n \times m$ (with $m$ substantially smaller than the original dimension $d$) via multiplication with a compression matrix $C$. We give a convex program for recovering the sparse component $S$ along with the compressed low-rank component $L \cdot C$, along with upper bounds on the error of this reconstruction that scales naturally with the compression dimension $m$ and coincides with existing results for the uncompressed setting $m = d$. Our results can also handle error introduced through additive noise or through missing data. The scaling of dimension, compression, and signal complexity in our theoretical results is verified empirically through simulations, and we also apply our method to a data set measuring chlorine concentration across a network of sensors to test its performance in practice.

## 1  Introduction

Principal component analysis (PCA) is a tool for providing a low-rank approximation to a data matrix $D \in \mathbb{R}^{n \times d}$, with the aim of reducing dimension or capturing the main directions of variation in the data. More recently, there has been increased focus on more general forms of PCA, that is more robust to realistic flaws in the data such as heavy-tailed outliers. The robust PCA (RPCA) problem formulates a decomposition of the data,

$$D \approx L + S ,$$

into a low-rank component $L$ (capturing trends across the data matrix) and a sparse component $S$ (capturing outlier measurements that may obscure the low-rank trends), which we seek to separate based only on observing the data matrix $D$ [3, 10]. Depending on the application, we may be primarily interested in one or the other component:

- In some settings, the sparse component $S$ may represent unwanted outliers, e.g. corrupted measurements—we may wish to clean the data by removing the outliers and recovering the low-rank component $L$.

- In other settings, the sparse component $S$ may contain the information of interest—for instance, in image or video data, $S$ may capture the foreground objects which are of interest, while $L$ may capture background components which we wish to subtract.

Existing methods to separate the sparse and low-rank components include convex [3, 10] and non-convex [9] methods, and can handle extensions or additional challenges such as missing data [3], column-sparse rather than elementwise-sparse structure [11], streaming data [6, 7], and different types of structures superimposed with a low-rank component [1].

In this paper, we examine the possibility of demixing sparse and low rank structure, under the additional challenge of working with data that has been compressed,

$$Y = D \cdot C \approx (L + S) \cdot C \in \mathbb{R}^{n \times m} \, ,$$

where $L, S \in \mathbb{R}^{n \times d}$ comprise the (approximately) low-rank and (approximately) sparse components of the original data matrix $D$, while $C \in \mathbb{R}^{d \times m}$ is a random or fixed compression matrix. In general, we think of the compression dimension $m$ as being significantly smaller than $d$, motivated by several considerations:

- Communication constraints: if the $n \times d$ data matrix consists of $d$-dimensional measurements taken at $n$ remote sensors, compression would allow the sensors to transmit information of dimension $m \ll d$;

- Storage constraints: storing a matrix with $nm$ many entries instead of $nd$ many entries;

- Data privacy: if the data is represented as the $n \times d$ matrix, where $n$-dimensional features were collected from $d$ individuals, we can preserve privacy by compressing the data by a random linear transformation and allow the access to database only through the compressed data. This privacy-preserving method has been called *matrix masking* in the privacy literature and studied by [12] in the context of high-dimensional linear regression.

Random projection methods have been shown to be highly useful for reducing dimensionality without much loss of accuracy for numerical tasks such as least squares regression [8] or low-rank matrix computations [5]. Here we use random projections to compress data while preserving the information about the underlying low-rank and sparse structure. [13] also applied random projection methods to the robust PCA problem, but their purpose is to accelerate the computational task of low-rank approximation, which is different from the aim of our work.

In the compressed robust PCA setting, we hope to learn about both the low-rank and sparse components. Unlike compressed sensing problems where sparse structure may be reconstructed perfectly with undersampling, here we face a different type of challenge:

- The sparse component $S$ is potentially identifiable from the compressed component $S \cdot C$, using the tools of compressed sensing; however,

- The low-rank component $L$ is *not* identifiable from its compression $L \cdot C$. Specifically, if we let $\mathcal{P}_C \in \mathbb{R}^{d \times d}$ be the projection operator onto the column span of $C$, then the two low-rank matrices $L$ and $L' = L \cdot \mathcal{P}_C$ cannot be distinguished after multiplication by $C$.

Therefore, our goal will be to recover both the sparse component $S$, and the *compressed* low-rank component $L \cdot C$. Note that recovering $L \cdot C$ is similar to the goal of recovering the column span of $L$, which may be a useful interpretation if we think of the columns of the data matrix $D$ as data points lying in $\mathbb{R}^n$; the column span of $L$ characterizes a low-rank subspace of $\mathbb{R}^n$ that captures the main trends in the data.

**Notation**   We will use the following notation throughout the paper. We write $[n] = \{1, \ldots, n\}$ for any $n \geq 1$. We write $\|v\|_0$ or $\|M\|_0$ to denote the number of nonzero entries in a vector $v$ or matrix $M$ (note that this is not in fact a norm). $M_{i*}$ denotes the $i$th row of a matrix $M$ and is treated as a column vector. We will use the matrix norms $\|M\|_{\mathsf{F}}$ (Frobenius norm), $\|M\|_1$ (elementwise $\ell_1$ norm), $\|M\|_\infty$ (elementwise $\ell_\infty$ norm), $\|M\|$ (spectral norm, i.e. largest singular value), and $\|M\|_*$ (nuclear norm, also known as the trace norm, given by the sum of the singular values of $M$).

## 2   Problem and method

We begin by formally defining the problem at hand. The data, which takes the form of a $n \times d$ matrix, is well-approximated by a sum $L^\star + S^\star$, where $L^\star$ is low-rank and $S^\star$ is sparse. However, we can only access this data through a (noisy) compression: our observed data is the $n \times m$ matrix

$$Y = (L^\star + S^\star) \cdot C + Z \, , \tag{1}$$

where $C \in \mathbb{R}^{d \times m}$ is the compression matrix, and $Z \in \mathbb{R}^{n \times m}$ absorbs all sources of error and noise—we discuss specific models for $Z$ later on.

Given this model, our goal will be to learn about both the low-rank and sparse structure. In the ordinary robust PCA setting, the task of separating the low-rank and sparse components has been known to be possible when the underlying low-rank component $L^\star$ satisfies certain conditions, e.g. incoherence condition in [3] or spikiness condition in [1]. In order to successfully decompose the low-rank and sparse component in the compressed data, we thus need the similar conditions to hold for the *compressed* low-rank component, which we define as the product $P^\star := L^\star \cdot C$. As we will see, if $L^\star$ satisfies the spikiness condition, i.e. $\|L^\star\|_\infty \le \alpha_0$, then the *compressed* low-rank component $P^\star$ satisfies the similar spikiness condition, i.e. a bound on $\|P^\star C\|_\infty$. This motivates the possibility to recover both the low-rank and sparse components in the case of compressed data.

As discussed above, while we can aim to recover the sparse component $S^\star$, there is no hope to recover the original low-rank component $L^\star$, since $L^\star$ is not identifiable in the compressed model. Therefore, we propose a natural convex program for recovering the underlying *compressed* low-rank component $P^\star = L^\star \cdot C$ and the sparse component $S^\star$. Note that as discussed in [5], random projection preserves the column span of $L^\star$, and so we can recover the column span of $L^\star$ via $P^\star$.

We define our estimators of the sparse component $S^\star$, and the low-rank product $P^\star$, as follows:

$$(\widehat{P}, \widehat{S}) = \underset{(P,S):\|PC^\top\|_\infty \le \alpha}{\arg\min} \left\{ \frac{1}{2}\|Y - P - S \cdot C\|_\mathsf{F}^2 + \nu\|P\|_* + \lambda\|S\|_1 \right\} . \tag{2}$$

Note that we impose the spikiness condition $\|PC^\top\|_\infty \le \alpha$ on $P$, in order to guarantee good performance for demixing such two superimposed components—in later section, we will see that the same condition holds for $P^\star$. This method is parametrized by the triple $(\alpha, \nu, \lambda)$, and natural scalings for these tuning parameters are discussed alongside our theoretical results.

## 2.1 Sources of errors and noise

Next, we give several examples of models and interpretations for the error term $Z$ in (1).

**Random noise** First, we may consider a model where the signal has an exact low-rank + sparse decomposition, with well-behaved additive noise added before and/or after the compression step:

$$Y = (L^\star + S^\star + Z_\mathsf{pre}) \cdot C + Z_\mathsf{post} ,$$

where the entries of the pre- and post-compression noise, $Z_\mathsf{pre}$ and $Z_\mathsf{post}$, are i.i.d. mean-zero sub-gaussian random variables. In this case, the noise term $Z$ in (1) is given by $Z = Z_\mathsf{pre} \cdot C + Z_\mathsf{post}$.

**Misspecified model** Next, we may consider a case where the original data can be closely approximated by a low-rank + sparse decomposition, but this decomposition is not exact. In this case, we could express the original (uncompressed) data as $L^\star + S^\star + Z_\mathsf{model}$, where $Z_\mathsf{model}$ captures the error of the low-rank + sparse decomposition. Then this model misspecification can be absorbed into the noise term $Z$, i.e. $Z = Z_\mathsf{model} \cdot C$.

**Missing data** Given an original data matrix $D = L^\star + S^\star$, we might have access only to a partial version of this matrix. We write $D_\Omega$ to denote the available data, where $\Omega \subset [n] \times [d]$ indexes the entries where data is available, and $(D_\Omega)_{ij} = D_{ij} \cdot \mathbb{1}_{ij \in \Omega}$. Then, a low-rank + sparse model for our compressed data is given by

$$Y = D_\Omega \cdot C = (L^\star + S_\Omega^\star) \cdot C + Z_\mathsf{missing} \cdot C ,$$

where $Z_\mathsf{missing} = L_\Omega^\star - L^\star$. In some settings, we may first want to adjust $D_\Omega$ before compressing the data, for instance, by reweighting the observed entries in $D_\Omega$ to ensure a closer approximation to $D$. Denoting the reweighted matrix of partial observations by $\widetilde{D}_\Omega$, we have compressed data

$$Y = \widetilde{D}_\Omega \cdot C = (L^\star + \widetilde{S}_\Omega^\star) \cdot C + Z_\mathsf{missing} \cdot C ,$$

with $Z_\mathsf{missing} = \widetilde{L}_\Omega^\star - L^\star$, and where $\widetilde{S}_\Omega^\star$ is the reweighted matrix of $S_\Omega^\star$. Then the error from the missing data can be absorbed into the $Z$ term, i.e. $Z = Z_\mathsf{missing} \cdot C$.

**Combinations** Finally, the observed data $Y$ may differ from the compressed low-rank + sparse decomposition $(L^\star + S^\star) \cdot C$ due to a combination of the factors above, in which case we may write

$$Z = (Z_\mathsf{pre} + Z_\mathsf{model} + Z_\mathsf{missing}) \cdot C + Z_\mathsf{post} .$$

## 2.2 Models for the compression matrix $C$

Next, we consider several scenarios for the compression matrix $C$.

**Random compression** In some settings, the original data naturally lies in $R^{n \times d}$, but is compressed by the user for some purpose. For instance, if we have data from $d$ individuals, with each data point lying in $\mathbb{R}^n$, we may compress this data for the purpose of providing privacy to the individuals in the data set. Alternately, we may compress data to adhere to constraints on communication bandwidth or on data storage. In either case, we control the choice of the compression matrix $C$, and are free to use a simple random model. Here we consider two models:

$$\text{Gaussian model: the entries of } C \text{ are generated as } C_{ij} \overset{\text{iid}}{\sim} N(0, 1/m). \tag{3}$$

$$\text{Orthogonal model: } C = \sqrt{d/m} \cdot U, \tag{4}$$
$$\text{where } U \in \mathbb{R}^{d \times m} \text{ is an orthonormal matrix chosen uniformly at random.}$$

Note that in each case, $\mathbb{E}\left[CC^\top\right] = \mathbf{I}_d$.

**Multivariate regression / multitask learning** In a multivariate linear regression, we observe a matrix of data $Y$ that follows a model $Y = X \cdot B + W$ where $X$ is an observed design matrix, $B$ is an unknown matrix of coefficients (generally the target parameter), and $W$ is a matrix of noise terms. Often, the rows of $Y$ are thought of as (independent) samples, where each row is a multivariate response. In this setting, the accuracy of the regression can often be improved by leveraging low-rank or sparse structure that arises naturally in the matrix of coefficients $B$. If $B$ is approximately low-rank + sparse, the methodology of this paper can be applied: taking the transpose of the multivariate regression model, we have $Y^\top = B^\top \cdot X^\top + W^\top$. Compare to our initial model (1), where we replace $Y$ with $Y^\top$, and use the compression matrix $C = X^\top$. Then, if $B^\top \approx L^\star + S^\star$ is a low-rank + sparse approximation, the multivariate regression can be formulated as a problem of the form (1) by setting the error term to equal $Z = (B^\top - L^\star - S^\star) \cdot X^\top + W^\top$.

## 3 Theoretical results

In this section, we develop theoretical error bounds for the compressed robust PCA problem under several of the scenarios described above. We first give a general deterministic result in Section 3.1, then specialize this result to handle scenarios of pre- and post-compression noise and missing data. Results for multivariate regression are given in the Supplementary Materials.

### 3.1 Deterministic result

We begin by stating a version of the Restricted Eigenvalue property found in the compressed sensing and sparse regression literature [2]:

**Definition 1.** For a matrix $X \in \mathbb{R}^{m \times d}$ and for $c_1, c_2 \geq 0$, $X$ satisfies the restricted eigenvalue property with constants $(c_1, c_2)$, denoted by $\mathsf{RE}_{m,d}(c_1, c_2)$, if

$$\|Xv\|_2 \geq c_1\|v\|_2 - c_2 \cdot \sqrt{\frac{\log(d)}{m}} \cdot \|v\|_1 \text{ for all } v \in \mathbb{R}^d . \tag{5}$$

We now give our main result for the accuracy of the convex program (2), a theorem that we will see can be specialized to many of the settings described earlier. This theorem gives a deterministic result and does not rely on a random model for the compression matrix $C$ or the error matrix $Z$.

**Theorem 1.** *Let* $L^\star \in \mathbb{R}^{n \times d}$ *be any matrix with* $\mathrm{rank}(L^\star) \leq r$, *and let* $S^\star \in \mathbb{R}^{n \times d}$ *be any matrix with at most* $s$ *nonzero entries per row, that is,* $\max_i \|S^\star_{i*}\|_0 \leq s$. *Let* $C \in \mathbb{R}^{d \times m}$ *be any compression matrix and define the data* $Y$ *and the error/noise term* $Z$ *as in* (1). *Let* $P^\star = L^\star \cdot C$ *as before. Suppose that* $C^\top$ *satisfies* $\mathsf{RE}_{m,d}(c_1, c_2)$, *where* $c_0 := c_1 - c_2 \cdot \sqrt{16s \log(d)/m} > 0$. *If parameters* $(\alpha, \nu, \lambda)$ *satisfy*

$$\alpha \geq \|L^\star CC^\top\|_\infty, \ \nu \geq 2\|Z\|, \ \lambda \geq 2\|ZC^\top\|_\infty + 4\alpha, \tag{6}$$

*then deterministically, the solution* $(\widehat{P}, \widehat{S})$ *to the convex program* (2) *satisfies*

$$\|\widehat{P} - P^\star\|_{\mathsf{F}}^2 + c_0^2\|\widehat{S} - S^\star\|_{\mathsf{F}}^2 \leq 18r\nu^2 + 9c_0^{-2}sn\lambda^2 .$$

We now highlight several applications of this theorem to specific settings: a random compression model with Gaussian or subgaussian noise, and a random compression model with missing data. (An application to the multivariate linear regression model is given in the Supplementary Materials.)

## 3.2 Results for random compression with subgaussian noise

Suppose compression matrix $C$ is random, and that the error term $Z$ in the model (1) comes from i.i.d. subgaussian noise, e.g. measurement error that takes place before and/or after the compression:

$$Z = Z_{\mathsf{pre}} \cdot C + Z_{\mathsf{post}} \ .$$

Our model for this setting is as follows: for fixed matrices $L^\star$ and $S^\star$, where $\mathrm{rank}(L^\star) \leq r$ and $\max_i \|S_{i*}^\star\|_0 \leq s$, we observe data

$$Y = (L^\star + S^\star + Z_{\mathsf{pre}}) \cdot C + Z_{\mathsf{post}} \ , \tag{7}$$

where the compression matrix $C$ is generated under either the Gaussian (3) or orthogonal (4) model, and where the noise matrices $Z_{\mathsf{pre}}, Z_{\mathsf{post}}$ are independent from each other and from $C$, with entries

$$(Z_{\mathsf{pre}})_{ij} \overset{\mathsf{iid}}{\sim} N(0, \sigma_{\mathsf{pre}}^2) \text{ and } (Z_{\mathsf{post}})_{ij} \overset{\mathsf{iid}}{\sim} N(0, \sigma_{\mathsf{post}}^2) \ .$$

For this section, we assume $d \geq m$ without further comment (that is, the compression should reduce the dimension of the data). Let $\sigma_{\mathsf{max}}^2 \geq \max\{\sigma_{\mathsf{pre}}^2, \sigma_{\mathsf{post}}^2\}$. Specializing the result of Theorem 1 to this setting, we obtain the following probablistic guarantee:

**Theorem 2.** *Assume the model* (7). *Suppose that* $\mathrm{rank}(L^\star) \leq r$, $\max_i \|S_{i*}^\star\|_0 \leq s$, *and* $\|L^\star\|_\infty \leq \alpha_0$. *Then there exist universal constants* $c, c', c'' > 0$ *such that if we define*

$$\alpha = 5\alpha_0 \sqrt{\frac{d \log(nd)}{m}}, \ \nu = 24\sigma_{\mathsf{max}} \sqrt{\frac{d(n+m)}{m}}, \ \lambda = 32\sigma_{\mathsf{max}} \sqrt{\frac{d \log(nd)}{m}} + 4\alpha \ ,$$

*and if* $m \geq c \cdot s \log(nd)$, *then the solution* $(\widehat{P}, \widehat{S})$ *to the convex program* (2) *satisfies*

$$\|\widehat{P} - P^\star\|_{\mathsf{F}}^2 + \|\widehat{S} - S^\star\|_{\mathsf{F}}^2 \leq c' \cdot \frac{d}{m} \left( \sigma_{\mathsf{max}}^2 \cdot r(n+m) + (\sigma_{\mathsf{max}}^2 + \alpha_0^2) \cdot sn \log(nd) \right) \tag{8}$$

*with probability at least* $1 - \frac{c''}{nd}$.

*Remark* 1. If the entries of $Z_{\mathsf{pre}}$ and $Z_{\mathsf{post}}$ are subgaussian rather than Gaussian, then the same result holds, except for a change in the constants appearing in the parameters $(\alpha, \nu, \lambda)$. (Recall that a random variable $X$ is $\sigma^2$-subgaussian if $\mathbb{E}\left[e^{tX}\right] \leq e^{t^2 \sigma^2 / 2}$ for all $t \in \mathbb{R}$.)

*Remark* 2. In the case $d = m$, our result matches Corollary 2 in Agarwal et al [1] exactly, except that our result involves multiplicative logarithm factor $\log(nd)$ in the $\alpha_0$ term whereas theirs does not.[1] This additional log factor arises when we upper bound $\|L^\star CC^\top\|_\infty$, which is unavoidable if we want the bound to hold with high probability.

*Remark* 3. Theorem 2 shows the natural scaling: the first term $r(n+m)$ is the degree of freedom for *compressed* rank $r$ matrix $P$ whereas the term $sn \log(nd)$ is the signal complexity of sparse component $S$, which has $sn$ many nonzero entries. The multiplicative factor $\frac{d}{m}\sigma_{\mathsf{max}}^2$ can be interpreted as the noise variance of the problem amplified by the compression.

## 3.3 Results for random compression with missing data

Next, we consider a missing data scenario where the original $n \times d$ matrix is only partially observed. The original (complete) data is $D = L^\star + S^\star \in \mathbb{R}^{n \times d}$, a low-rank + sparse decomposition.[2] However, only a subset $\Omega \subset [n] \times [d]$ of entries are observed—we are given access to $D_{ij}$ for each $(i, j) \in \Omega$. After a reweighting step, we compress this data with a compression matrix $C \in \mathbb{R}^{d \times m}$, for instance, in order to reduce communication, storage, or computation requirements.

First, we specify a model for the missing data. For each $(i,j) \in [n] \times [d]$, let $\rho_{ij} \in [0,1]$ be the probability that this entry is *observed*. Additionally, we assume that the sampling scheme is independent across all entries, and that the $\rho_{ij}$'s are known.[3]

To proceed, we first define a reweighted version of the partially observed data matrix and then multiply by the compression matrix $C$:

$$Y = \widetilde{D}_\Omega \cdot C \quad \text{where} \quad (\widetilde{D}_\Omega)_{ij} = D_{ij}/\rho_{ij} \cdot \mathbb{1}_{ij \in \Omega} \ . \tag{9}$$

Define also the reweighted versions of the low rank and sparse components,

$$(\widetilde{L}^\star_\Omega)_{ij} = L_{ij}/\rho_{ij} \cdot \mathbb{1}_{ij \in \Omega} \quad \text{and} \quad (\widetilde{S}^\star_\Omega)_{ij} = S_{ij}/\rho_{ij} \cdot \mathbb{1}_{ij \in \Omega} \ ,$$

and note that we then have

$$Y = \left(\widetilde{L}^\star_\Omega + \widetilde{S}^\star_\Omega\right) \cdot C = \left(L^\star + \widetilde{S}^\star_\Omega\right) \cdot C + Z \ , \tag{10}$$

where $Z = (\widetilde{L}^\star_\Omega - L^\star) \cdot C$. The role of the reweighting step (9) is to ensure that this noise term $Z$ has mean zero. Note that in the reformulation (10) of the model, $Y$ is approximated with a compression of $L^\star + \widetilde{S}^\star_\Omega$, where $L^\star$ is the original low rank component while $\widetilde{S}^\star_\Omega$ is defined above. While the original sparse component $S^\star$, is not identifiable via the missing data model (since we have no information to help us recover entries $S^\star_{ij}$ for $(i,j) \notin \Omega$), this new decomposition $L^\star + \widetilde{S}^\star_\Omega$ now has a sparse component that *is* identifiable, since by definition, $\widetilde{S}^\star_\Omega$ preserves the sparsity of $S^\star$ but has no nonzero entries in unobserved locations, that is, $(\widetilde{S}^\star_\Omega)_{ij} = 0$ whenever $(i,j) \notin \Omega$.

With this model in place, we obtain the following probabilistic guarantee for this setting, which is another specialized version of Theorem 1. We note that we again have no assumptions on the values of the entries in $S^\star$, only on the sparsity level—e.g. there is no bound assumed on $\|S^\star\|_\infty$.

**Theorem 3.** *Assume the model* (9). *Suppose that* $\mathrm{rank}(L^\star) \leq r$, $\max_i \|S^\star_{i*}\|_0 \leq s$, *and* $\|L^\star\|_\infty \leq \alpha_0$. *If the sampling scheme satisfies* $\rho_{ij} \geq \rho_{\mathsf{min}}$ *for all* $(i,j) \in [n] \times [d]$ *for some positive constant* $\rho_{\mathsf{min}} > 0$, *then there exist universal constants* $c, c', c'' > 0$ *such that if we define*

$$\alpha = 5\alpha_0 \sqrt{\frac{d\log(nd)}{m}}, \ \nu = 10\rho_{\mathsf{min}}^{-1}\alpha_0 \sqrt{\frac{d(n+m)\log(nd)}{m}}, \ \lambda = 12\rho_{\mathsf{min}}^{-1}\alpha_0\sqrt{\frac{d\log^2(nd)}{m}} + 4\alpha \ ,$$

*and if* $m \geq c \cdot s\log(nd)$, *then the solution* $(\widehat{P}, \widehat{S})$ *to the convex program* (2) *satisfies*

$$\|\widehat{P} - P^\star\|_{\mathsf{F}}^2 + \|\widehat{S} - \widetilde{S}^\star_\Omega\|_{\mathsf{F}}^2 \leq c' \cdot \frac{d}{m} \cdot \rho_{\mathsf{min}}^{-2}\alpha_0^2 \left(r(n+m)\log(nd) + sn\log^2(nd)\right)$$

*with probability at least* $1 - \frac{c''}{nd}$.

## 4 Experiments

In this section, we first use simulated data to study the behavior of the convex program (2) for different compression dimensions, signal complexities and missing levels, which show the close agreement with the scaling predicted by our theory. We also apply our method to a data set consisting of chlorine measurements across a network of sensors. For simplicity, in all experiments, we select $\alpha = \infty$, which is easier for optimization and generally results in a solution that still has low spikiness (that is, the solution is the same as if we had imposed a bound with finite $\alpha$).

### 4.1 Simulated data

Here we run a series of simulations on compressed data to examine the performance of the convex program (2). In all cases, we used the compression matrix $C$ generated under the orthogonal model (4). We solve the convex program (2) via alternating minimization over $L$ and $S$, selecting the regularization parameters $\nu$ and $\lambda$ that minimizes the squared Frobenius error. All results are averaged over 5 trials.

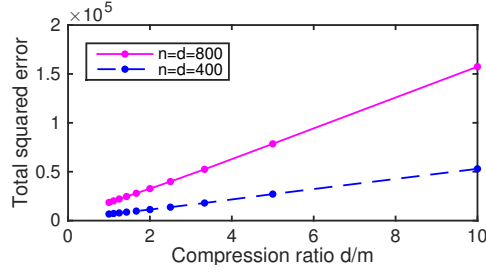

Figure 1: Results for the noisy data experiment. The total squared error, calculated as in Theorem 2, is plotted against the compression ratio $d/m$. Note the linear scaling, as predicted by the theory.

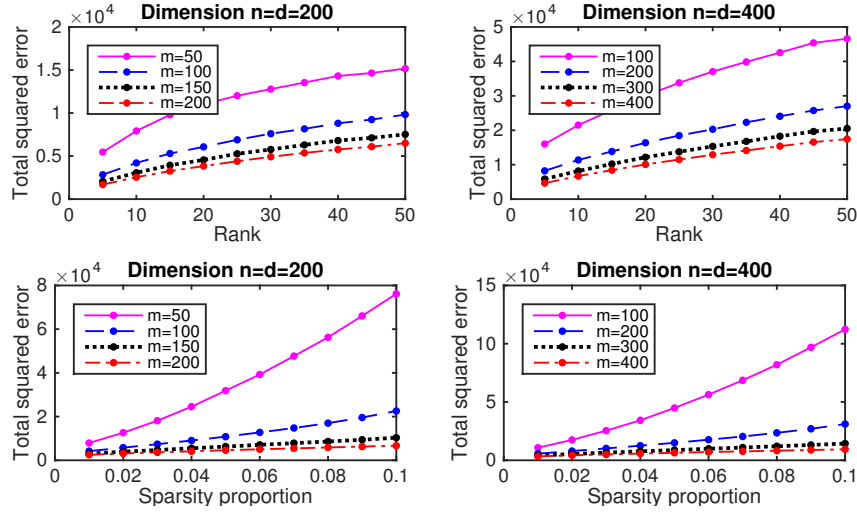

Figure 2: Results for the varying-rank (top row) and varying-sparsity (bottom row) experiments. The total squared error, calculated as in Theorem 2, is plotted against the rank $r$ or sparsity proportion $s/d$. Note the nearly linear scaling for most values of $m$.

**Simulation 1: compression ratio.**  First we examine the role of the compression dimension $m$. We fix the matrix dimension $n = d \in \{400, 800\}$. The low-rank component is given by $L^\star = \sqrt{r} \cdot U V^\top$, where $U$ and $V$ are $n \times r$ and $d \times r$ matrices with i.i.d. $N(0,1)$ entries, for rank $r = 10$. The sparse component $S^\star$ has 1% of its entries generated as $5 \cdot N(0,1)$, that is, $s = 0.01d$. The data is $D = L^\star + S^\star + Z$, where $Z_{ij} \overset{\text{iid}}{\sim} N(0, 0.25)$. Figure 1 shows the squared Frobenius error $\|\widehat{P} - P^\star\|_F^2 + \|\widehat{S} - S^\star\|_F^2$ plotted against the compression ratio $d/m$. We see error scaling linearly with the compression ratio, which supports our theoretical results.

**Simulation 2: rank and sparsity.**  Next we study the role of rank and sparsity, for a matrix of size $n = d = 200$ or $n = d = 400$. We generate the data $D$ as before, but we either vary the rank $r \in \{5, 10, \dots, 50\}$, or we vary the sparsity $s$ with $s/d \in \{0.01, 0.02, \dots, 0.1\}$. Figure 2 shows the squared Frobenius error plotted against either the varying rank or the varying sparsity. We repeat this experiment for several different compression dimensions $m$. We see a little deviation from linear scaling for the smallest $m$, which can be due to the fact that our theorems give upper bounds rather than tight matching upper and lower bounds (or perhaps the smallest value of $m$ does not satisfy the condition stated in the theorems). However, for all but the smallest $m$, we see error scaling nearly linearly with rank or with sparsity, which is consistent with our theory.

**Simulation 3: missing data.**  Finally, we perform experiments under the existence of missing entries in the data matrix $D = L^\star + S^\star$. We fix dimensions $n = d = 400$ and generate $L^\star$ and $S^\star$ as before, with $r = 10$ and $s = 0.01d$, but do not add noise. To introduce the missing entries in the data, we use a uniform sampling scheme, where each entry of $D$ is observed with probability $\rho$,

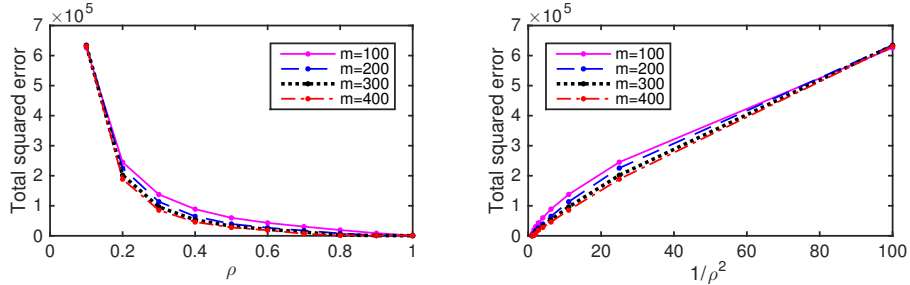

Figure 3: Results for the missing data experiment. The total squared error, calculated as in Theorem 3, is plotted against $\rho$ (proportion of observed data) or against $1/\rho^2$, for various values of $m$, based on one trial. Note the nearly linear scaling with respect to $1/\rho^2$.

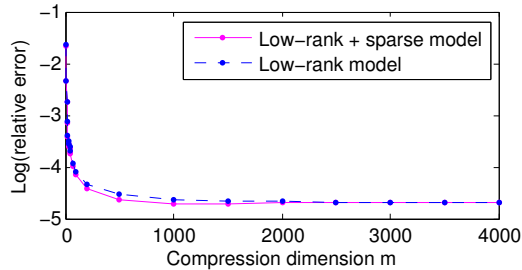

Figure 4: Results for the chlorine data (averaged over 2 trials), plotting the log of the relative error on the test set for a low-rank + sparse model and a low-rank-only model. The low-rank + sparse model performs better across a range of compression dimensions $m$ (up to 8–9% reduction in error).

with $\rho \in \{0.1, 0.2, \ldots, 1\}$. Figure 3 shows the squared Frobenius error $\|\widehat{P} - P^\star\|_F^2 + \|\widehat{S} - \widetilde{S}_\Omega^\star\|_F^2$ (see Theorem 3 for details) across a range of probabilities $\rho$. We see that the squared error scales approximately linearly with $1/\rho^2$, as predicted by our theory.

## 4.2 Chlorine sensor data

To illustrate the application of our method to a specific application, we consider chlorine concentration data from a network of sensors.[4] The data contains a realistic simulation of chlorine concentration measurements from $n = 166$ sensors in a hydraulic system over $d = 4310$ time points. We assume $D$ is well approximated with a low-rank + sparse decomposition. We then compress the data using the orthogonal model (4) and study the performance of our estimators (2) for varying $m$. In order to evaluate performance, we use 80% of the entries to fit the model, 10% as a validation set for selecting tuning parameters, and the final 10% as a test set. We compare against a low-rank matrix reconstruction, equivalent to setting $\widehat{S} = 0$ and fitting only the low-rank component $\widehat{L}$. (Details are given in the Supplementary Materials.) The results are displayed in Figure 4, where we see that the error of the recovery grows smoothly with compression dimension $m$, and that the low-rank + sparse decomposition gives better data reconstruction than the low-rank-only model.

## 5 Discussion

In this paper, we have examined the robust PCA problem under data compression, where we seek to decompose a data matrix into low-rank + sparse components with access only to a partial projection of the data. This provides a tool for accurate modeling of data with multiple superimposed structures, while enabling restrictions on communication, privacy, or other considerations that may make compression necessary. Our theoretical results show an intuitive tradeoff between the compression ratio and the error of the fitted low-rank + sparse decomposition, which coincides with existing results in the extreme case of no compression (compression ratio = 1). Future directions for this problem include adapting the method to the streaming data (online learning) setting.

## Footnotes

[1]Note that $s \cdot n$ in our paper is equivalent to $s$ in [1], since their work defines $s$ to be the total number of nonzero entries in $S^\star$ while we count entries per row.

[2]For clarity of presentation, we do not include additive noise before or after compression in this section. However, our theoretical analysis for additive noise (Theorem 2) and for missing data (Theorem 3) can be combined in a straightforward way to obtain an error bound scaling as a sum of the two respective bounds.

[3] In practice, the assumption that $\rho_{ij}$'s are known is not prohibitive. For example, we might model $\rho_{ij} = \alpha_i\beta_j$ (the row and column locations of the observed entries are chosen independently, e.g. see [4]), or a logistic model, $\log\left(\frac{\rho_{ij}}{1-\rho_{ij}}\right) = \alpha_i + \beta_j$. In either case, fitting a model using the observed set $\Omega$ is extremely accurate.

[4]Data obtained from `http://www.cs.cmu.edu/afs/cs/project/spirit-1/www/`

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
