[Supplementary Material]

# Supplementary materials for:
# Robust PCA with compressed data

**Wooseok Ha**
University of Chicago
haywse@uchicago.edu

**Rina Foygel Barber**
University of Chicago
rina@uchicago.edu

## A   Results for multivariate linear regression

In multivariate regression / multitask learning, we are given a collection of $n$ regression problems in $\mathbb{R}^m$, where the data at hand is of the form $Y_j \in \mathbb{R}^m$ for each task $j = 1, \cdots, n$, and $X \in \mathbb{R}^{m \times d}$ is the design matrix. If we denote the unknown regression matrix as $B \in \mathbb{R}^{d \times n}$, our observed data can be written as

$$Y = X \cdot B + W$$

where $W \in \mathbb{R}^{m \times n}$ is the observation noise matrix. Suppose the matrix of coefficients $B$ has an exact low-rank plus sparse decomposition, in which case we can write $B^\top = L^\star + S^\star$ where $\text{rank}(L^\star) \le r$ and $\max_i \|S^\star_{i*}\|_0 \le s$. Then, by taking the transpose of the observation model

$$Y^\top = (L^\star + S^\star) \cdot X^\top + W^\top \ , \tag{1}$$

we see that this is a special instance of our initial model (1), where compression matrix is given by $C = X^\top$ and error term is $Z = W^\top$. We emphasize here that in most multivariate regression / multitask learning problems, compression matrix $X^\top$ cannot be controlled by the user and is obtained through observing the specific event or phenomenon. For this section, we assume a fixed design matrix $X$ satisfying a restricted eigenvalue property (a standard assumption for many sparse regression problems [2]). In this setting, we aim to recover the low-rank product $P^\star = L^\star \cdot X^\top$ and the sparse matrix $S^\star$ by solving the convex program (2). We state the performance of the recovered solution, which can be obtained from Theorem 1.

**Proposition 1.** *Assume the model* (1)*, where* $B^\top = L^\star + S^\star$*, with* $\text{rank}(L^\star) \le r$*, and* $\max_i\|S^\star_{i*}\|_0 \le s$*. Assume that* $X$ *has unit-length columns and satisfies* $\mathsf{RE}_{m,d}(c_1, c_2)$*, where*

$$c_0 := c_1 - c_2 \cdot \sqrt{\frac{16 s \log(d)}{m}} > 0,$$

*and assume that* $W_{ij} \overset{\text{iid}}{\sim} N(0, \sigma^2)$*. Then there exist universal constants* $c, c', c'' > 0$ *such that if we choose*[1]

$$\alpha \ge \|L^\star X^\top X\|_\infty, \ \nu \ge 6\sigma\sqrt{n+m}, \ \lambda \ge 4\sigma\sqrt{\log(nd)} + 4\alpha \ ,$$

*and if* $m \ge c \cdot s \log(nd)$*, then the solution* $(\widehat{P}, \widehat{S})$ *to the convex program* (2) *(with* $Y^\top$ *in place of* $Y$*) satisfies*

$$\|\widehat{P} - P^\star\|_{\mathsf{F}}^2 + \|\widehat{S} - S^\star\|_{\mathsf{F}}^2 \le c' \cdot c_0^{-4} \cdot \left(r\sigma^2(n+m) + (\sigma^2\log(nd) + \alpha^2)sn\right)$$

*with probability at least* $1 - \frac{c''}{nd}$*.*

*Remark* 1. If the entries of $W$ are subgaussian rather than Gaussian, then the same result holds, except for a change in the constants appearing in the parameters $(\alpha, \nu, \lambda)$.

*Remark* 2. Corollary 4 in Agarwal et al [1] provides the result that leads to the same scaling as our result. However, they essentially require $m \geq d$ as opposed to our case that only requires $m \geq c \cdot s \log(nd)$. We also provide recovery guarantee of the compressed low-rank component $P^\star = L^\star \cdot X^\top$ rather than the low-rank component of coefficients $L^\star$ as in [1].

## B  Details for chlorine data experiment

This section gives details for the chlorine data experiment described in Section 4.2. We first introduce missing entries in the data matrix $D$ and focus on predicting those entries by solving the convex program (2) based on the reweighted partially observed data matrix $D_\Omega$. To do this, we follow a missing at random model, partitioning the $nd$ entries into three sets, $\Omega_{\text{obs}} \cup \Omega_{\text{val}} \cup \Omega_{\text{test}}$, where $\Omega_{\text{obs}}$ contains the observed data (80% of all entries), $\Omega_{\text{val}}$ is a validation set for tuning (10% of all entries), and $\Omega_{\text{test}}$ is the test set (10% of all entries). For each $j \in [d]$, we define the set of observed entries in column $j$ as $O_j := \{i \in [n] : (i,j) \in \Omega_{\text{obs}}\}$ and similarly for $V_j$ and $T_j$. We will then proceed as follows: Fixing $\nu$ and $\lambda$, solve the convex program (2) using the compressed reweighted data matrix $\widetilde{D}_{\Omega_{\text{obs}}} \cdot C$ to obtain an estimator $(\widehat{P}, \widehat{S})$. According to the low-rank + sparse decomposition, the column span of $\widehat{P}$ should closely approximate $D - \rho \cdot \widehat{S}$, and so we should have $(D - \rho \cdot \widehat{S})_{*j} \approx \widehat{P} \cdot w_j$ for some $w_j$. We compute $w_j$ using least squares on the observed data, i.e.

$$w_j = (\widehat{P}_{O_j *}^\top \widehat{P}_{O_j *})^{-1} \widehat{P}_{O_j *} (D - \rho \cdot \widehat{S})_{O_j, j} .$$

We then predict the missing entries located at the validation set entries $V_j$ as

$$\widehat{D}_{V_j, j} = \widehat{P}_{V_j *} \cdot w_j + \widehat{S}_{V_j, j} .$$

Since we have no information for recovering $S^\star_{ij}$ for $(i,j) \notin \Omega_{\text{obs}}$, we remove the largest 1% of entries of prediction error $\widehat{D}_{\Omega_{\text{val}}} - D_{\Omega_{\text{val}}}$ and evaluate the prediction performance only using the remaining 99% of the entries:

$$\|(\widehat{D}_{\Omega_{\text{val}}} - D_{\Omega_{\text{val}}})_{\text{bottom 99%}}\|_{\mathsf{F}}^2 .$$

With this prediction performance criteria, we choose the optimal regularization parameters $\nu_{\text{opt}}, \lambda_{\text{opt}}$ and finally outputs the prediction error evaluated on the test set:

$$\|(\widehat{D}_{\Omega_{\text{test}}} - D_{\Omega_{\text{test}}})_{\text{bottom 99%}}\|_{\mathsf{F}}^2 .$$

## C  Proofs of theorems

### C.1  Proof of Theorem 1

**Background**  First we introduce a few definitions which are now standard for the analysis of this type of structured problem, using the decomposability of the $\ell_1$ norm and the nuclear norm. Let $\Omega \subset [n] \times [d]$ be the support of the true sparse component $S^\star$, and let $\Omega_i \subset [d]$ be the $i$-th row of $\Omega$, i.e. $\Omega_i = \{j : S^\star_{ij} \neq 0\}$. Let $T$ be the tangent space to the nuclear norm at $P^\star$, which is given by [3]

$$T = \{AV^\top + UB^\top : \text{ any matrices } A \in \mathbb{R}^{n \times r}, B \in \mathbb{R}^{m \times r}\} ,$$

where $P^\star = U\Sigma V^\top$ is a singular value decomposition of $P^\star$ with $U \in \mathbb{R}^{n \times r}$ and $V \in \mathbb{R}^{m \times r}$. It is known [2] that, for any $S \in \mathbb{R}^{n \times d}$, for each row $i \in [n]$,

$$\|S^\star_{i*}\|_1 - \|S_{i*}\|_1 \leq \|(S - S^\star)_{i\Omega_i}\|_1 - \|(S - S^\star)_{i\Omega_i^c}\|_1 , \tag{2}$$

which trivially yields

$$\|S^\star\|_1 - \|S\|_1 \leq \|\mathcal{P}_\Omega (S - S^\star)\|_1 - \|\mathcal{P}_\Omega^\perp (S - S^\star)\|_1 , \tag{3}$$

where $\mathcal{P}_\Omega ()$ and $\mathcal{P}_\Omega^\perp ()$ denote projection onto the subspace of matrices supported on $\Omega$, and onto the orthogonal subspace. Furthermore for any $P \in \mathbb{R}^{n \times m}$,

$$\|P^\star\|_* - \|P\|_* \leq \|\mathcal{P}_T (P - P^\star)\|_* - \|\mathcal{P}_T^\perp (P - P^\star)\|_* , \tag{4}$$

where $\mathcal{P}_T ()$ and $\mathcal{P}_T^\perp ()$ denote projection onto the subspace $T \subset \mathbb{R}^{n \times m}$, and onto its orthogonal complement $T^\perp$. Throughout, we will use the facts that $\|M\|_* \leq \|\mathcal{P}_T (M)\|_* + \|\mathcal{P}_T^\perp (M)\|_*$ and similarly $\|M\|_1 \leq \|\mathcal{P}_\Omega (M)\|_1 + \|\mathcal{P}_\Omega^\perp (M)\|_1$ without comment.

| Notation | Definition | Name |
|---|---|---|
| $[n]$ | $\{1, \dots, n\}$ | |
| $\|v\|_0$ or $\|M\|_0$ | $\sum_i \mathbb{1}_{v_i \neq 0}$ or $\sum_{ij} \mathbb{1}_{M_{ij} \neq 0}$ | number of nonzeros in vector $v$ or matrix $M$ |
| $M_{*j}$ | $(M_{1j}, M_{2j}, \dots)$ | $j$th column of the matrix $M$ |
| $M_{i*}$ | $(M_{i1}, M_{i2}, \dots)$ | $i$th row of the matrix $M$ (always treated as a column vector) |
| $M_{AB}$ | $(M_{ij})_{i \in A, j \in B}$ | the submatrix of $M$ indexed by $A \times B$ |
| $\|M\|_{\mathsf{F}}$ | $\sqrt{\sum_{ij} M_{ij}^2}$ | Frobenius norm |
| $\|M\|_1$ | $\sum_{ij} |M_{ij}|$ | elementwise $\ell_1$ norm |
| $\|M\|_\infty$ | $\max_{ij} |M_{ij}|$ | elementwise $\ell_\infty$ norm |
| $\|M\|_{2,\infty}$ | $\max_i \|M_{i*}\|_2$ | largest row $\ell_2$ norm |
| $\|M\|$ | $\max_v \frac{\|Mv\|_2}{\|v\|_2}$ | spectral norm (a.k.a. operator norm) |
| $\|M\|_*$ | $\sum_i \sigma_i(M)$ where $\sigma_1(M), \sigma_2(M), \dots$ are the singular values of $M$ | nuclear norm (a.k.a. trace norm) |
| $c, c', c'', \dots$ | | universal constants (whose definitions may change from one result to another) |

Table 1: Notation used throughout the proofs.

**Basic inequality** By optimality,

$$\frac{1}{2}\|Y - \widehat{P} - \widehat{S}C\|_{\mathsf{F}}^2 + \nu\|\widehat{P}\|_* + \lambda\|\widehat{S}\|_1 \leq \frac{1}{2}\|Y - P^\star - S^\star G\|_{\mathsf{F}}^2 + \nu\|P^\star\|_* + \lambda\|S^\star\|_1 . \quad (5)$$

Define errors $\Delta^P = \widehat{P} - P^\star$ and $\Delta^S = \widehat{S} - S^\star$. Using our model (1) for $Y$, and applying (3) and (4), we rearrange terms to obtain

$$\frac{1}{2}\|\Delta^P + \Delta^S C\|_{\mathsf{F}}^2 \leq \langle Z, \Delta^P + \Delta^S C\rangle + \nu\left(\|\mathcal{P}_T\left(\Delta^P\right)\|_* - \|\mathcal{P}_T^\perp\left(\Delta^P\right)\|_*\right)$$
$$+ \lambda\left(\|\mathcal{P}_\Omega\left(\Delta^S\right)\|_1 - \|\mathcal{P}_\Omega^\perp\left(\Delta^S\right)\|_1\right)$$
$$\leq \|Z\| \cdot \|\Delta^P\|_* + \|ZC^\top\|_\infty \cdot \|\Delta^S\|_1 + \nu\left(\|\mathcal{P}_T\left(\Delta^P\right)\|_* - \|\mathcal{P}_T^\perp\left(\Delta^P\right)\|_*\right)$$
$$+ \lambda\left(\|\mathcal{P}_\Omega\left(\Delta^S\right)\|_1 - \|\mathcal{P}_\Omega^\perp\left(\Delta^S\right)\|_1\right)$$
$$\leq \|\mathcal{P}_T\left(\Delta^P\right)\|_*(\nu + \|Z\|) - \|\mathcal{P}_T^\perp\left(\Delta^P\right)\|_*(\nu - \|Z\|)$$
$$+ \|\mathcal{P}_\Omega\left(\Delta^S\right)\|_1(\lambda + \|ZC^\top\|_\infty) - \|\mathcal{P}_\Omega^\perp\left(\Delta^S\right)\|_1(\lambda - \|ZC^\top\|_\infty) .$$

Now we consider the left-hand side. We have

$$\frac{1}{2}\|\Delta^P + \Delta^S C\|_{\mathsf{F}}^2 = \frac{1}{2}\|\Delta^P\|_{\mathsf{F}}^2 + \frac{1}{2}\|\Delta^S C\|_{\mathsf{F}}^2 + \langle \Delta^P, \Delta^S C\rangle$$
$$\geq \frac{1}{2}\|\Delta^P\|_{\mathsf{F}}^2 + \frac{1}{2}\|\Delta^S C\|_{\mathsf{F}}^2 - \|\Delta^P C^\top\|_\infty \cdot \|\Delta^S\|_1$$
$$\geq \frac{1}{2}\|\Delta^P\|_{\mathsf{F}}^2 + \frac{1}{2}\|\Delta^S C\|_{\mathsf{F}}^2 - 2\alpha\|\Delta^S\|_1 ,$$

where the last step uses $\|\Delta^P C^\top\|_\infty \leq \|\widehat{P}C^\top\|_\infty + \|P^\star C^\top\|_\infty \leq 2\alpha$ by the assumption $\|P^\star C^\top\|_\infty \leq \alpha$ (6) and the constraint $\|\widehat{P}C^\top\|_\infty \leq \alpha$ in the optimization problem (2). Including this into the work above, then,

$$\frac{1}{2}\|\Delta^P\|_{\mathsf{F}}^2 + \frac{1}{2}\|\Delta^S C\|_{\mathsf{F}}^2 \leq \|\mathcal{P}_T\left(\Delta^P\right)\|_*(\nu + \|Z\|) - \|\mathcal{P}_T^\perp\left(\Delta^P\right)\|_*(\nu - \|Z\|)$$
$$+ \|\mathcal{P}_\Omega\left(\Delta^S\right)\|_1(\lambda + \|ZC^\top\|_\infty + 2\alpha) - \|\mathcal{P}_\Omega^\perp\left(\Delta^S\right)\|_1(\lambda - \|ZC^\top\|_\infty - 2\alpha)$$
$$\leq \nu(1.5\|\mathcal{P}_T\left(\Delta^P\right)\|_* - 0.5\|\mathcal{P}_T^\perp\left(\Delta^P\right)\|_*) + \lambda(1.5\|\mathcal{P}_\Omega\left(\Delta^S\right)\|_1 - 0.5\|\mathcal{P}_\Omega^\perp\left(\Delta^S\right)\|_1) , \quad (6)$$

where the last step uses the assumptions (6) on the parameters $(\alpha, \nu, \lambda)$.

**Strong convexity**  Next, we need to use the restricted strong convexity assumption on $C$. First, we consider the rows of $\widehat{S}$ individually. Fixing $\widehat{P}$, we note that the optimization problem (2) separates over the rows of $\widehat{S}$: ignoring the term $\nu\|\widehat{P}\|_*$ which is constant with respect to $S$, we have

$$\frac{1}{2}\|Y - \widehat{P} - \widehat{S}C\|_\mathsf{F}^2 + \lambda\|\widehat{S}\|_1 = \sum_i \left(\frac{1}{2}\|Y_{i*} - \widehat{P}_{i*} - C^\top \widehat{S}_{i*}\|_2^2 + \lambda\|\widehat{S}_{i*}\|_1\right) .$$

Therefore, $\widehat{S}_{i*}$ is the minimizer of the term in parentheses, for each $i$, and in particular we have

$$\frac{1}{2}\|Y_{i*} - \widehat{P}_{i*} - C^\top \widehat{S}_{i*}\|_2^2 + \lambda\|\widehat{S}_{i*}\|_1 \leq \frac{1}{2}\|Y_{i*} - \widehat{P}_{i*} - C^\top S_{i*}^\star\|_2^2 + \lambda\|S_{i*}^\star\|_1 .$$

Rearranging terms and applying (2), we get

$$\frac{1}{2}\|C^\top(\widehat{S}_{i*} - S_{i*}^\star)\|_2^2 \leq \langle Y_{i*} - \widehat{P}_{i*} - C^\top S_{i*}^\star, \Delta_{i*}^S\rangle + \lambda\left(\|\Delta_{i\Omega_i}^S\|_1 - \|\Delta_{i\Omega_i^c}^S\|_1\right)$$

$$\leq \|C(Y_{i*} - \widehat{P}_{i*} - C^\top S_{i*}^\star)\|_\infty \cdot \|\Delta_{i*}^S\|_1 + \lambda\left(\|\Delta_{i\Omega_i}^S\|_1 - \|\Delta_{i\Omega_i^c}^S\|_1\right) .$$

We also have

$$\|C(Y_{i*} - \widehat{P}_{i*} - C^\top S_{i*}^\star)\|_\infty = \|C(Z_{i*} - (\widehat{P} - P^\star)_{i*})\|_\infty \leq \|(Z - (\widehat{P} - P^\star))C^\top\|_\infty$$

$$\leq \|ZC^\top\|_\infty + \|\widehat{P}C^\top\|_\infty + \|P^\star C^\top\|_\infty \leq \|ZC^\top\|_\infty + 2\alpha \leq \lambda/2 ,$$

by the assumption (6) on $\lambda$. Combining this with the above, we then have

$$\frac{1}{2}\|C^\top\Delta_{i*}^S\|_2^2 \leq \lambda\left(1.5\|\Delta_{i\Omega_i}^S\|_1 - 0.5\|\Delta_{i\Omega_i^c}^S\|_1\right) ,$$

and since the left-hand side is nonnegative, we therefore have

$$\|\Delta_{i\Omega_i^c}^S\|_1 \leq 3\|\Delta_{i\Omega_i}^S\|_1 ,$$

that is, for *every* row of the sparse matrix, a substantial portion of the $\ell_1$ norm of the error is located on the correct support. Therefore,

$$\|\Delta_{i*}^S\|_1 = \|\Delta_{i\Omega_i^c}^S\|_1 + \|\Delta_{i\Omega_i}^S\|_1 \leq 4\|\Delta_{i\Omega_i}^S\|_1 \leq 4\sqrt{s}\|\Delta_{i\Omega_i}^S\|_2 \leq 4\sqrt{s}\|\Delta_{i*}^S\|_2 ,$$

where the next-to-last inequality holds because $|\Omega_i| \leq s$ by assumption on the sparsity of the row $S_{i*}^\star$.

Next, by assumption of the theorem, $C^\top$ satisfies $\mathsf{RE}_{m,d}(c_1, c_2)$. We then have

$$\|C^\top\Delta_{i*}^S\|_2 \geq c_1\|\Delta_{i*}^S\|_2 - c_2\cdot\sqrt{\frac{\log(d)}{m}}\|\Delta_{i*}^S\|_1 \geq \left(c_1 - c_2\cdot 4\sqrt{s}\cdot\sqrt{\frac{\log(d)}{m}}\right)\|\Delta_{i*}^S\|_2 = c_0\|\Delta_{i*}^S\|_2 ,$$

where the last step uses the definition of $c_0$ in the theorem. (Recall that $c_0 > 0$ by assumption.) Summing over the rows, we then have

$$\|\Delta^S C\|_\mathsf{F}^2 = \sum_i \|C^\top\Delta_{i*}^S\|^2 \geq \sum_i c_0^2\|\Delta_{i*}^S\|_2^2 = c_0^2\|\Delta^S\|_\mathsf{F}^2 . \tag{7}$$

**Combining everything**  Now we return to (6) and plug in our result in (7), to obtain

$$\frac{1}{2}\|\Delta^P\|_\mathsf{F}^2 + \frac{c_0^2}{2}\|\Delta^S\|_\mathsf{F}^2 \leq \nu(1.5\|\mathcal{P}_T\left(\Delta^P\right)\|_* - 0.5\|\mathcal{P}_T^\perp\left(\Delta^P\right)\|_*) + \lambda(1.5\|\mathcal{P}_\Omega\left(\Delta^S\right)\|_1 - 0.5\|\mathcal{P}_\Omega^\perp\left(\Delta^S\right)\|_1) .$$

Removing negative terms from the right-hand side and multiplying by 2,

$$\|\Delta^P\|_\mathsf{F}^2 + c_0^2\|\Delta^S\|_\mathsf{F}^2 \leq 3\nu\|\mathcal{P}_T\left(\Delta^P\right)\|_* + 3\lambda\|\mathcal{P}_\Omega\left(\Delta^S\right)\|_1 .$$

Since $\mathrm{rank}(\mathcal{P}_T\left(\Delta^P\right)) \leq 2r$ by definition of $T$, and similarly since $\|\mathcal{P}_\Omega\left(\Delta^S\right)\|_0 \leq sn$ by definition of $\Omega$, we have

$$\|\Delta^P\|_\mathsf{F}^2 + c_0^2\|\Delta^S\|_\mathsf{F}^2 \leq 3\nu\|\mathcal{P}_T\left(\Delta^P\right)\|_\mathsf{F}\cdot\sqrt{2r} + 3\lambda\|\mathcal{P}_\Omega\left(\Delta^S\right)\|_\mathsf{F}\cdot\sqrt{sn}$$

$$\leq 3\nu\|\Delta^P\|_\mathsf{F}\cdot\sqrt{2r} + 3\lambda\|\Delta^S\|_\mathsf{F}\cdot\sqrt{sn}$$

$$\leq \sqrt{\|\Delta^P\|_\mathsf{F}^2 + c_0^2\|\Delta^S\|_\mathsf{F}^2}\cdot\sqrt{18r\nu^2 + 9c_0^{-2}sn\lambda^2} ,$$

where the last step uses the Cauchy-Schwarz inequality. In particular, this implies that

$$\|\Delta^P\|_\mathsf{F}^2 + c_0^2\|\Delta^S\|_\mathsf{F}^2 \leq 18r\nu^2 + 9c_0^{-2}sn\lambda^2 ,$$

which proves the desired result.

## C.2 Proof of Theorem 2

This result is a straightforward application of Theorem 1. It will be sufficient to check that, with the stated probability, the following statements all hold:

$$C^\top \text{ satisfies } \mathsf{RE}_{m,d}(c_1, c_2), \text{ with } c_0 := c_1 - c_2\sqrt{\frac{16s\log(d)}{m}} > 0, \tag{8}$$

and

$$\alpha \geq \|L^\star CC^\top\|_\infty, \ \nu \geq 2\|Z\|, \ \lambda \geq 2\|ZC^\top\|_\infty + 4\alpha. \tag{9}$$

To prove that (8) holds, the following lemma is sufficient (along with the assumption $m \geq c \cdot s\log(nd)$):

**Lemma 1.** *Under either the Gaussian model* (3) *or the orthogonal model* (4) *for the compression matrix* $C$, *for any* $\delta > 0$, $C^\top$ *satisfies* $\mathsf{RE}_{m,d}(c_1, c_2)$ *for constants*

$$c_1 = \frac{1}{4(2 + \sqrt{2})} \quad and \quad c_2 = \frac{9}{2 + \sqrt{2}}$$

*with probability at least* $1 - c'e^{-cm}$, *where* $c, c' > 0$ *are universal constants.*

To prove (9), we consider the first inequality by treating $L^\star$ as fixed and analysing the random model for $C$:

**Lemma 2.** *Under either the Gaussian model* (3) *or the orthogonal model* (4) *for* $C$, *for any fixed matrix* $L^\star \in \mathbb{R}^{n \times d}$ *and fixed* $\delta > 0$, *if* $m \geq 16\log(nd)$, *then*

$$\mathbb{P}\left\{\|L^\star CC^\top\|_\infty > \|L^\star\|_\infty \cdot \left(1 + \sqrt{\frac{16d\log(nd)}{m}}\right)\right\} \leq \frac{4}{nd}.$$

For the second and third inequalities in (9), we first have the following bound on $C$:

**Lemma 3.** *Under either the Gaussian* (3) *or orthogonal* (4) *model for* $C$, *with probability at least* $1 - 2de^{-m/8}$,

$$\|C\| \leq \sqrt{12d/m} \text{ and } \|C\|_{2,\infty} \leq 2.$$

Next, we consider $C$ as fixed and analyse the random model for the noise terms $Z_{\mathsf{pre}}$ and $Z_{\mathsf{post}}$ (we can treat $C$ as fixed since the noise is generated independently from $C$). Fixing $C$, the rows of $Z = Z_{\mathsf{pre}}C + Z_{\mathsf{post}}$ are i.i.d. draws from the distribution $N(0, \sigma_{\mathsf{pre}}^2 C^\top C + \sigma_{\mathsf{post}}^2 \mathbf{I}_m)$. Then, writing $\Sigma = \sigma_{\mathsf{pre}}^2 C^\top C + \sigma_{\mathsf{post}}^2 \mathbf{I}_m$, we have

$$\|Z\| \leq \|Z \cdot \Sigma^{-1/2}\| \cdot \sqrt{\|\Sigma\|} \leq 3\sqrt{n + m} \cdot \sqrt{\|\Sigma\|},$$

with probability at least $1 - e^{-m}$, where the last step uses the fact that $Z \cdot \Sigma^{-1/2}$ is a $n \times m$ matrix with i.i.d. standard normal entries, and applies Davidson and Szarek [4, Theorem II.13]. Furthermore,

$$\|\Sigma\| \leq \sigma_{\mathsf{pre}}^2 \|C\|^2 + \sigma_{\mathsf{post}}^2 \leq \sigma_{\mathsf{max}}^2 \cdot (12d/m + 1) \leq \left(4\sigma_{\mathsf{max}}\sqrt{d/m}\right)^2,$$

where the last step follows from Lemma 3. Combining these steps,

$$\|Z\| \leq 12\sigma_{\mathsf{max}}\sqrt{n + m} \cdot \sqrt{d/m}.$$

Next, we need to bound $\|ZC^\top\|_\infty$. Note that the entries are distributed as

$$(ZC^\top)_{ij} \sim N(0, \sigma_{\mathsf{pre}}^2 (CC^\top CC^\top)_{jj} + \sigma_{\mathsf{post}}^2 (CC^\top)_{jj}),$$

and this variance term is bounded as

$$\sigma_{\mathsf{pre}}^2 (CC^\top CC^\top)_{jj} + \sigma_{\mathsf{post}}^2 (CC^\top)_{jj} = \sigma_{\mathsf{pre}}^2 \mathbf{e}_j^\top CC^\top CC^\top \mathbf{e}_j + \sigma_{\mathsf{post}}^2 \mathbf{e}_j^\top CC^\top \mathbf{e}_j$$

$$\leq \sigma_{\mathsf{max}}^2 \left(\|C\|^2 + 1\right) \|C^\top \mathbf{e}_j\|_2^2 \leq \sigma_{\mathsf{max}}^2 \cdot (12d/m + 1) \cdot 2^2 \leq \left(8\sigma_{\mathsf{max}}\sqrt{d/m}\right)^2,$$

where the last step follows from Lemma 3. Therefore, using standard tail bounds on the normal distribution, with probability at least $1 - \frac{2}{nd}$,

$$\|ZC^\top\|_\infty = \max_{ij} \left|(ZC^\top)_{ij}\right| \leq 8\sigma_{\mathsf{max}}\sqrt{d/m} \cdot 2\sqrt{\log(nd)}.$$

## C.3 Proof of Theorem 3

This result is another immediate consequence of Theorem 1, with $\widetilde{S}_\Omega^\star$ in place of $S^\star$ (note that $\max_i \|(\widetilde{S}_\Omega^\star)_{i*}\|_0 \leq \max_i \|S_{i*}^\star\|_0 \leq s$ by assumption) . Since the restricted eigenvalue property and the condition $\alpha \geq \|L^\star C C^\top\|_\infty$ follow from Lemma 1 and Lemma 2 respectively, it is sufficient to check that, with the stated probability, the following statements both hold:

$$\nu \geq 2\|Z\|, \ \lambda \geq 2\|ZC^\top\|_\infty + 4\alpha, \tag{10}$$

where $Z = (\widetilde{L}_\Omega^\star - L^\star) \cdot C$ as defined before. Let $B_{ij} \overset{\perp\!\!\!\perp}{\sim} \mathsf{Bernoulli}(\rho_{ij})$ be an indicator variable for $(i,j) \in \Omega$, that is, for whether we observe entry $(i,j)$. Then we can write $\widetilde{L}_\Omega^\star$ as

$$(\widetilde{L}_\Omega^\star)_{ij} = \frac{B_{ij}}{\rho_{ij}} \cdot L_{ij}^\star$$

for each $(i,j) \in [n] \times [d]$, and so $Z$ can be written as

$$Z = \sum_{ij} \left( \frac{B_{ij}}{\rho_{ij}} - 1 \right) \cdot E_{ij} \tag{11}$$

where $E_{ij} = L_{ij}^\star \cdot \mathbf{e}_i C_{j*}^\top \in \mathbb{R}^{n \times m}$, and where $\mathbf{e}_i \in \mathbb{R}^n$ is the $i$-th standard basis vector and $C_{j*} \in \mathbb{R}^m$ is $j$-th row of the compression matrix $C$. To prove the first inequality in (10), we consider $C$ as fixed and analyse the random model for $B_{ij}$'s. We first have the following bound on the sum of random scalars times fixed matrices:

**Lemma 4** (Adapted from Tropp [8, Theorem 4.1.1]). *Let $A_1, \ldots, A_L \in \mathbb{R}^{d_1 \times d_2}$ be fixed matrices, and let $B_1, \ldots, B_L$ be independent mean-zero random variables, such that for each $\ell = 1, \ldots, L$, $B_\ell$ is $\sigma^2$-subgaussian, that is,*

$$\mathbb{E}\left[ e^{tB_\ell} \right] \leq e^{\sigma^2 t^2 / 2} \text{ for all } t \in \mathbb{R} \,.$$

*Then*

$$\mathbb{P}\left\{ \left\| \sum_{\ell=1}^L B_\ell A_\ell \right\| \geq t \right\} \leq (d_1 + d_2) \exp \left\{ -\frac{t^2}{2\sigma^2 \max\left\{ \|\sum_{\ell=1}^L A_\ell A_\ell^\top\|, \|\sum_{\ell=1}^L A_\ell^\top A_\ell\| \right\}} \right\} \,.$$

To apply Lemma 4 to the error term expression $Z$ in (11), we first show that the random scalar, defined by

$$\widetilde{B}_{ij} = \frac{B_{ij}}{\rho_{ij}} - 1 \,,$$

is $\sigma^2$-subgaussian with $\sigma^2 = 2\mu^2$ for all $(i,j) \in [n] \times [d]$. To see this, first note that $\mathbb{E}\left[\widetilde{B}_{ij}\right] = 0$ and $|\widetilde{B}_{ij}|$ is bounded by $\mu$ for all $(i,j) \in [n] \times [d]$. If $|t| \geq (2\mu)^{-1}$, then

$$\mathbb{E}\left[ e^{t\widetilde{B}_{ij}} \right] \leq \mathbb{E}\left[ e^{(2\mu^2 t^2 + \widetilde{B}_{ij}^2 / 2\mu^2)/2} \right] = e^{\mu^2 t^2} \mathbb{E}\left[ e^{\widetilde{B}_{ij}^2 / 4\mu^2} \right] \leq e^{\mu^2 t^2} e^{1/4} \leq e^{2\mu^2 t^2}$$

where the last inequality holds due to $|t| \geq (2\mu)^{-1}$. If $|t| \leq (2\mu)^{-1}$, we have $|t\widetilde{B}_{ij}| \leq 1/2$, and so

$$\mathbb{E}\left[ e^{t\widetilde{B}_{ij}} \right] \leq 1 + t\mathbb{E}\left[\widetilde{B}_{ij}\right] + t^2 \mathbb{E}\left[\widetilde{B}_{ij}^2\right] = 1 + t^2 \mathbb{E}\left[\widetilde{B}_{ij}^2\right] \leq e^{t^2 \mathbb{E}[\widetilde{B}_{ij}^2]} \leq e^{\mu^2 t^2}$$

where the first inequality follows from the fact that $e^x \leq 1 + x + x^2$ for $|x| \leq 1/2$. Therefore, we apply Lemma 4 to the error term expression (11) so that, with probability at least $1 - \frac{1}{nd}$ (with respect to the randomness of the $B_{ij}$'s),

$$\|Z\| \leq \sqrt{4\mu^2 \max\left\{ \|\sum_{ij} E_{ij} E_{ij}^\top\|, \|\sum_{ij} E_{ij}^\top E_{ij}\| \right\} \log\left(nd \cdot (n+m)\right)} \,.$$

Next, we derive the probabilistic bound on $\max\left\{ \|\sum_{ij} E_{ij} E_{ij}^\top\|, \|\sum_{ij} E_{ij}^\top E_{ij}\| \right\}$. We first state the following bound on $C$:

**Lemma 5.** *Under either the Gaussian* (3) *or orthogonal* (4) *model for* $C$, *with probability at least* $1 - 2e^{-m}$,

$$\|C\| \leq \sqrt{12d/m} \text{ and } \|C\|_{\mathsf{F}} \leq \sqrt{3d} .$$

Direct calculation shows that

$$\|\sum_{ij} E_{ij} E_{ij}^{\top}\| = \max_{i} \left( \sum_{j=1}^{d} \|C_{j*}\|_2^2 L_{ij}^{\star}{}^2 \right) \leq \alpha_0^2 \cdot \|C\|_{\mathsf{F}}^2$$

and

$$\|\sum_{ij} E_{ij}^{\top} E_{ij}\| = \|\sum_{ij} L_{ij}^{\star}{}^2 C_{j*} C_{j*}^{\top}\| \leq \alpha_0^2 \cdot n\|C\|^2 .$$

Then, applying Lemma 5, with probability at least $1 - 2e^{-m}$,

$$\max \left\{ \|\sum_{ij} E_{ij} E_{ij}^{\top}\|, \|\sum_{ij} E_{ij}^{\top} E_{ij}\| \right\} \leq \alpha_0^2 \max \left\{ \|C\|_{\mathsf{F}}^2, n\|C\|^2 \right\} \leq \alpha_0^2 \cdot 12\frac{d(n+m)}{m} .$$

In total, we have with probability at least $1 - \frac{2}{nd}$,

$$\|Z\| \leq \mu\alpha_0 \sqrt{48\frac{d(n+m)}{m} \log\left(nd(n+m)\right)} .$$

Since $m \leq d$, we can write $\log(nd(n+m)) \leq \log(nd(n+d)) \leq \max\{\log(2n^2d), \log(2nd^2)\} \leq 2\log(nd)$, where we assume $n, d \geq 2$ to avoid triviality. So,

$$\|Z\| \leq 10\mu\alpha_0 \sqrt{\frac{d(n+m)}{m} \log(nd)} . \tag{12}$$

Next, we need to bound on $\|ZC^{\top}\|_{\infty}$. Note that

$$\|ZC^{\top}\|_{\infty} = \|(\widetilde{L}_{\Omega}^{\star} - L^{\star})CC^{\top}\|_{\infty} \leq \|\widetilde{L}_{\Omega}^{\star} - L^{\star}\|_{\infty} + \|(\widetilde{L}_{\Omega}^{\star} - L^{\star})(CC^{\top} - \mathbf{I}_d)\|_{\infty} .$$

By our assumptions, we can immediately bound $\|\widetilde{L}_{\Omega}^{\star} - L^{\star}\|_{\infty} \leq \mu\alpha_0$. Next consider the term $\|(\widetilde{L}_{\Omega}^{\star} - L^{\star})(CC^{\top} - \mathbf{I}_d)\|_{\infty}$. We first consider $C$ as fixed and analyse the random model for $B_{ij}$'s. The $(i, \ell)$-th entry of $(\widetilde{L}_{\Omega}^{\star} - L^{\star})(CC^{\top} - \mathbf{I}_d)$ can be written as

$$\left[ (\widetilde{L}_{\Omega}^{\star} - L^{\star})(CC^{\top} - \mathbf{I}_d) \right]_{i\ell} = \sum_{j} \widetilde{B}_{ij} \cdot L_{ij}^{\star}(CC^{\top} - \mathbf{I}_d)_{j\ell} ,$$

which is mean zero random scalar and bounded above by $\mu\alpha_0\|CC^{\top} - \mathbf{I}_d\|_{\infty}$. Therefore, applying Hoeffding's Lemma and union bound, with probability at least $1 - \frac{1}{nd}$ (with respect to the randomness of the $B_{ij}$'s),

$$\|(\widetilde{L}_{\Omega}^{\star} - L^{\star})(CC^{\top} - \mathbf{I}_d)\|_{\infty} \leq \sqrt{2d(\alpha_0 + \alpha_1)^2\mu^2\|CC^{\top} - \mathbf{I}_d\|_{\infty}^2 \log\left(2n^2d^2\right)} . \tag{13}$$

For the bound on $\|CC^{\top} - \mathbf{I}_d\|_{\infty}$, we have the following result:

**Lemma 6.** *Under either the Gaussian* (3) *or orthogonal* (4) *model for* $C$, *with probability at least* $1 - \frac{4}{nd}$,

$$\|CC^{\top} - \mathbf{I}_d\|_{\infty} \leq \sqrt{\frac{24\log(nd)}{m}} .$$

Combining (13) with Lemma 6, we have with probability at least $1 - \frac{5}{nd}$,

$$\|ZC^{\top}\|_{\infty} \leq 7\mu\alpha_0 \sqrt{\frac{d\log(nd)\log(2n^2d^2)}{m}} \leq 12\mu\alpha_0 \sqrt{\frac{d\log^2(nd)}{m}} .$$

## C.4 Proof of Proposition 1

In order to apply Theorem 1, it is sufficient to check the conditions

$$\alpha \geq \|L^\star CC^\top\|_\infty = \|L^\star X^\top X\|_\infty, \ \nu \geq 2\|Z\| = 2\|W^\top\|, \ \lambda \geq 2\|ZC^\top\|_\infty + 4\alpha = 2\|W^\top X\|_\infty + 4\alpha.$$

The lower bound on $\alpha$ is true by definition. For the lower bound on $\nu$, the following result from Davidson and Szarek [4, Theorem II.13]: if entries of $W \in \mathbb{R}^{m \times n}$ are generated as i.i.d. $N(0, \sigma^2)$, then with probability at least $1 - e^{-m}$,

$$\|W\| \leq 3\sigma\sqrt{n+m} \ .$$

Finally, we bound $\|W^\top X\|_\infty$. The entries of $W^\top X$ are distributed as

$$(W^\top X)_{ij} \sim N(0, \sigma^2\|X_{*j}\|_2^2) = N(0, \sigma^2) \ ,$$

where the last step holds since the columns of $X$ are normalized. Therefore, using the standard tail bounds on the normal distribution, with probability at least $1 - \frac{2}{nd}$,

$$\|W^\top X\|_\infty = \max_{ij} |(W^\top X)_{ij}| \leq 2\sigma\sqrt{\log(nd)} \ .$$

# D  Proofs of lemmas

## D.1  Concentration lemma

We first state a concentration result under the Gaussian model (3) or the orthogonal model (4):

**Lemma 7.** *Under either the Gaussian model* (3) *or the orthogonal model* (4), *for any fixed vector $w \in \mathbb{R}^d$ and any $\epsilon > 0$,*

$$\mathbb{P}\left\{\frac{\|C^\top w\|_2^2}{\|w\|_2^2} - 1 > \epsilon\right\} \leq \exp\left\{-\frac{m}{8} \cdot \min\{\epsilon, \epsilon^2\}\right\}, \mathbb{P}\left\{\frac{\|C^\top w\|_2^2}{\|w\|_2^2} - 1 < -\epsilon\right\} \leq \exp\left\{-\frac{m}{4}\epsilon^2\right\} \ . \tag{14}$$

*Proof.* Under the Gaussian model,

$$m \cdot \frac{\|C^\top w\|_2^2}{\|w\|_2^2} \sim \chi_m^2$$

and therefore, by the $\chi^2$ tail bounds of Laurent and Massart [6, Lemma 1], for any $t > 0$,

$$\mathbb{P}\left\{m \cdot \frac{\|C^\top w\|_2^2}{\|w\|_2^2} > m + 2\sqrt{mt} + 2t\right\} \leq e^{-t} \ \text{and} \ \mathbb{P}\left\{m \cdot \frac{\|C^\top w\|_2^2}{\|w\|_2^2} < m - 2\sqrt{mt}\right\} \leq e^{-t} \ .$$

Setting $t = \frac{m}{8} \cdot \min\{\epsilon, \epsilon^2\}$, we obtain the desired result (14). Next, turning to the orthogonal model, we have $G = \sqrt{\frac{d}{m}} \cdot U$ where $U \in \mathbb{R}^{d \times m}$ is an orthonormal matrix chosen uniformly at random. Let $v \in \mathbb{R}^d$ be a random unit vector. Then $\|U^\top w\|_2^2$ is equal in distribution to $v_1^2 + \cdots + v_m^2$. In this setting, Dubhashi and Panconesi [5, Lemma 2.4] states that, for any $0 < \beta_0 < 1$,

$$\mathbb{P}\left\{v_1^2 + \cdots + v_m^2 < \beta_0\frac{m}{d}\right\} \leq \exp\left\{\frac{m}{2}\left(1 - \beta_0 + \log(\beta_0)\right)\right\}$$

and for any $\beta_1 > 1$,

$$\mathbb{P}\left\{v_1^2 + \cdots + v_m^2 > \beta_1\frac{m}{d}\right\} \leq \exp\left\{\frac{m}{2}\left(1 - \beta_1 + \log(\beta_1)\right)\right\} \ .$$

Next, set $\beta_1 = 1 + \epsilon$. Then, since for all $x > 0$ we have $\log(1 + x) \leq x - \frac{\min\{x, x^2\}}{4}$, then

$$1 - \beta_1 + \log(\beta_1) \leq 1 - (1 + \epsilon) + \epsilon - \frac{\min\{\epsilon, \epsilon^2\}}{4} = -\frac{\min\{\epsilon, \epsilon^2\}}{4} \ .$$

Therefore,

$$\mathbb{P}\left\{\frac{\|C^\top w\|_2^2}{\|w\|_2^2} > 1 + \epsilon\right\} \leq \exp\left\{-\frac{m}{8} \cdot \min\{\epsilon, \epsilon^2\}\right\} \ .$$

Next we want to bound the probability of the event $\frac{\|C^\top w\|_2^2}{\|w\|_2^2} < 1 - \epsilon$. If $\epsilon \geq 1$ then trivially this cannot occur. If instead $\epsilon < 1$, then we set $\beta_0 = 1 - \epsilon$. Since $\log(1-x) \leq -x - \frac{x^2}{2}$ for all $0 < x < 1$, we have

$$1 - \beta_0 + \log(\beta_0) = 1 - (1-\epsilon) - \epsilon - \frac{\epsilon^2}{2} = -\frac{\epsilon^2}{2} \,,$$

and so

$$\mathbb{P}\left\{\frac{\|C^\top w\|_2^2}{\|w\|_2^2} < 1 - \epsilon\right\} \leq \exp\left\{-\frac{m}{4} \cdot \epsilon^2\right\} \,.$$

This is sufficient to prove the desired bound. $\qquad\square$

## D.2 Proof of Lemma 2 (elementwise bounds)

Set $\epsilon = \sqrt{\frac{16 \log(nd)}{m}}$ and note that $\epsilon \leq 1$ by assumption. For each $i \in [n]$, define the unit vector $v_i = \frac{L_{i*}^\star}{\|L_{i*}^\star\|_2}$ (treated as a column vector). Now fix any $i \in [n]$ and any $j \in [d]$. Then

$$\left(L^\star C C^\top\right)_{ij} = \|L_{i*}^\star\|_2 \cdot v_i^\top C C^\top \mathbf{e}_j = \|L_{i*}^\star\|_2 \cdot \frac{1}{4}\left(\|C^\top(v_i + \mathbf{e}_j)\|_2^2 - \|C^\top(v_i - \mathbf{e}_j)\|_2^2\right) \,.$$

By Lemma 7, with probability at least $1 - 4e^{-m\epsilon^2/8}$,

$$\left|\frac{\|C^\top(v_i + \mathbf{e}_j)\|_2^2}{\|v_i + \mathbf{e}_j\|_2^2} - 1\right| \leq \epsilon \quad \text{and} \quad \left|\frac{\|C^\top(v_i - \mathbf{e}_j)\|_2^2}{\|v_i - \mathbf{e}_j\|_2^2} - 1\right| \leq \epsilon \,.$$

If these bounds hold, then

$$\begin{aligned}
\left(L^\star C C^\top\right)_{ij} &= \|L_{i*}^\star\|_2 \cdot \frac{1}{4}\left(\|C^\top(v_i + \mathbf{e}_j)\|_2^2 - \|C^\top(v_i - \mathbf{e}_j)\|_2^2\right) \\
&\leq \|L_{i*}^\star\|_2 \cdot \frac{1}{4}\left((1+\epsilon) \cdot \|v_i + \mathbf{e}_j\|_2^2 - (1-\epsilon) \cdot \|v_i - \mathbf{e}_j\|_2^2\right) \\
&= \|L_{i*}^\star\|_2 \cdot \frac{1}{4}\left(\left(\|v_i + \mathbf{e}_j\|_2^2 - \|v_i - \mathbf{e}_j\|_2^2\right) + \epsilon\left(\|v_i + \mathbf{e}_j\|_2^2 + \|v_i - \mathbf{e}_j\|_2^2\right)\right) \\
&= \|L_{i*}^\star\|_2 \cdot \frac{1}{4}\left(4\langle v_i, \mathbf{e}_j\rangle + \epsilon\left(2\|v_i\|_2^2 + 2\|\mathbf{e}_j\|_2^2\right)\right) \\
&= \|L_{i*}^\star\|_2 \cdot \left(\langle v_i, \mathbf{e}_j\rangle + \epsilon\right) \quad \text{since } \|v_i\|_2 = \|\mathbf{e}_j\|_2 = 1 \\
&= \langle L_{i*}^\star, \mathbf{e}_j\rangle + \epsilon\|L_{i*}^\star\|_2 \quad \text{by definition of } v_i \\
&= L_{ij}^\star + \epsilon\|L_{i*}^\star\|_2 \\
&\leq \|L^\star\|_\infty\left(1 + \epsilon\sqrt{d}\right) \,.
\end{aligned}$$

Using the same arguments, the same bound holds for $-(L^\star C C^\top)_{ij}$, and therefore,

$$\left|(L^\star C C^\top)_{ij}\right| \leq \|L^\star\|_\infty\left(1 + \epsilon\sqrt{d}\right) \,.$$

Applying the union bound over each $i \in [n]$ and each $j \in [d]$, we see that

$$\|L^\star C C^\top\|_\infty \leq \|L^\star\|_\infty(1 + \epsilon\sqrt{d})$$

with probability at least

$$1 - nd \cdot 4e^{-m\epsilon^2/8} = 1 - 4nd\exp\left\{\frac{m}{8}\left(\sqrt{\frac{16\log(nd)}{m}}\right)^2\right\} = 1 - \frac{4}{nd} \,.$$

## D.3 Proof of Lemma 3 (bounds on random compression matrix)

First we treat $\|C\|$. Under the orthogonal model, $\|C\| \leq \sqrt{d/m}$ trivially, while under the Gaussian model for $C$ (3), $\|C\| \leq \sqrt{d/m}(2 + \sqrt{2})$ with probability at least $1 - e^{-m}$ by again applying [4, Theorem II.13]. Next consider $\|C\|_{2,\infty} = \max_{i=1,\dots,d}\|C^\top \mathbf{e}_i\|_2$. For each $i$, by Lemma 7,

$$\mathbb{P}\left\{\|C^\top \mathbf{e}_j\|_2 > 2\right\} \leq e^{-m/8} \,.$$

Therefore,

$$\mathbb{P}\left\{\|C\|_{2,\infty} > 2\right\} \leq d \cdot e^{-m/8} \,.$$

### D.4 Proof of Lemma 4 (bounds on matrix subgaussian series)

Tropp [8, Theorem 4.1.1] proves this exact statement for the special case that either $B_\ell \overset{\text{iid}}{\sim} N(0,1)$ (Gaussian variables) or $B_\ell \overset{\text{iid}}{\sim} \{\pm 1\}$ (Rademacher variables). To see why the statement holds in this more general case, we observe that for Corollary 4.2 in Tropp, the distribution of the $B_\ell$'s is used only once: to prove the bound

$$\mathbb{E}\left[e^{tB_\ell A}\right] \preceq e^{t^2 A^2/2}$$

for each $\ell$ and for any fixed Hermitian matrix $A$. For the general case, take a fixed Hermitian matrix $A$, with $A = Q\Lambda Q^\top$ its eigendecomposition. We have

$$\begin{aligned}
\mathbb{E}\left[e^{tB_\ell A}\right] &= \mathbb{E}\left[e^{Q\cdot(tB_\ell\Lambda)\cdot Q^\top}\right] \\
&= Q \cdot \text{diag}\{\mathbb{E}\left[e^{tB_\ell\lambda_i}\right]\} \cdot Q^\top \\
&\preceq Q \cdot \text{diag}\{e^{\sigma^2 t^2 \lambda_i^2/2}\} \cdot Q^\top \\
&= e^{Q\cdot(\sigma^2 t^2 \Lambda^2/2)\cdot Q^\top} \\
&= e^{\sigma^2 t^2 A^2/2} \ .
\end{aligned}$$

Therefore, this is sufficient to see that Corollary 4.2 of Tropp holds in this case also.

### D.5 Proof of Lemma 5 (bounds on random compression matrix)

The result for $\|C\|$ follows from Lemma 3. Next consider $\|C\|_{\mathsf{F}}^2$. Under the orthogonal model, $\|C\|_{\mathsf{F}}^2 = \text{tr}(C^\top C) = d$ holds. Under the Gaussian model for $C$, we note that $\|C\|_{\mathsf{F}}^2 \sim \chi_{md}^2/m$. By the $\chi^2$ tail bounds of Laurent and Massart [6, Lemma 1], we have

$$\mathbb{P}\left\{\|C\|_{\mathsf{F}}^2 \geq d + 2\sqrt{d} + 2\right\} \leq e^{-m} \ .$$

Since $3d \geq d + 2\sqrt{d} + 2$ for $d \geq 1$, with probability at least $1 - e^{-m}$, we have $\|C\|_{\mathsf{F}}^2 \leq 3d$.

### D.6 Proof of Lemma 6 (bounds on random compression matrix)

This result is the consequence of Lemma 7 and union bound. Set $\epsilon = \sqrt{\frac{24\log(nd)}{m}}$ and $\epsilon \leq 1$. By Lemma 7, with probability at least $1 - 2e^{-m\epsilon^2/8}$, for $i \neq j$,

$$(CC^\top - \mathbf{I}_d)_{ij} = \mathbf{e}_i^\top(CC^\top - \mathbf{I}_d)\mathbf{e}_j = \frac{1}{4}(\|C^\top(\mathbf{e}_i + \mathbf{e}_j)\|^2 - \|C^\top(\mathbf{e}_i - \mathbf{e}_j)\|^2)$$

$$\leq \frac{1}{4}((1+\epsilon)\|\mathbf{e}_i + \mathbf{e}_j\|^2 - (1-\epsilon)\|\mathbf{e}_i - \mathbf{e}_j\|^2) \leq \epsilon \ .$$

The same bound holds for $-(CC^\top - \mathbf{I}_d)_{ij}$ if we use the same arguments, and so with probability at least $1 - 4e^{-m\epsilon^2/8}$,

$$|(CC^\top - \mathbf{I}_d)_{ij}| \leq \epsilon \ .$$

For $i = j$, applying Lemma 7 again, with probability at least $1 - 2e^{-m\epsilon^2/8}$,

$$|(CC^\top - \mathbf{I}_d)_{ij}| = |\mathbf{e}_i^\top(CC^\top - \mathbf{I}_d)\mathbf{e}_j| = |\|C^\top\mathbf{e}_j\|_2^2 - 1| \leq \epsilon \ .$$

Applying the union bound over each $(i,j) \in [d] \times [d]$, we have that

$$\|CC^\top - \mathbf{I}_d\|_\infty \leq \sqrt{\frac{24\log(nd)}{m}}$$

with probability at least $1 - 4d^2 e^{-m\epsilon^2/8} \geq 1 - \frac{4}{nd}$.

### D.7 Proof of Lemma 1 (restricted strong convexity)

First, for the Gaussian model (3), by Raskutti et al. [7, Theorem 1], for universal constants $c, c' > 0$,

$$\mathbb{P}\left\{\|C^\top x\|_2 \geq \frac{1}{4}\|x\|_2 - 9\sqrt{\frac{\log(d)}{m}}\|x\|_1 \text{ for all } x \in \mathbb{R}\right\} \geq 1 - c'e^{-cm} .$$

Next, we turn to the orthogonal model (4). Let $H \in \mathbb{R}^{d \times m}$ be a matrix with $H_{ij} \overset{\text{iid}}{\sim} N(0, 1/m)$, let $H = UDV^\top$ be its singular value decomposition, and without loss of generality take $C = \sqrt{\frac{d}{m}} \cdot U$ (since $H$ is rotation invariant and so $U$ is uniformly distributed over the space of uniform matrices, this satisfies the orthogonal model (4)). Then for any $x \in \mathbb{R}^d$,

$$\|H^\top x\|_2^2 = \|VDU^\top x\|_2^2 \leq \|VD\|^2\|U^\top x\|_2^2 = \|H\|^2 \cdot \frac{m}{d} \cdot \|C^\top x\|_2^2 .$$

By the work above for the Gaussian model, with probability at least $1 - c'e^{-cm}$,

$$\|H^\top x\|_2 \geq \frac{1}{4}\|x\|_2 - 9\sqrt{\frac{\log(d)}{m}}\|x\|_1 \text{ for all } x \in \mathbb{R}^d ,$$

and by Davidson and Szarek [4, Theorem II.13], with probability at least $1 - e^{-m}$,

$$\|H\| \leq \sqrt{\frac{d}{m}} + 1 + \sqrt{\frac{2m}{d}} \leq \sqrt{\frac{d}{m}}\left(2 + \sqrt{2}\right) .$$

Combining all these bounds, with probability at least $1 - c'e^{-cm} - e^{-m} \geq 1 - (c'+1)e^{-\min\{c,1\}\cdot m}$, for all $x \in \mathbb{R}^d$,

$$\|C^\top x\|_2 \geq \frac{1}{4(2 + \sqrt{2})}\|x\|_2 - \frac{9}{2 + \sqrt{2}}\sqrt{\frac{\log(d)}{m}}\|x\|_1 .$$

Clearly, this statement holds also for the Gaussian model as well (since this is a strictly weaker result than the one stated above.)

## Footnotes

[1]Requiring that we choose $\alpha$ to satisfy $\alpha \ge \|L^\star X^\top X\|_\infty$ is of course nontrivial since $L^\star$ is not known. However, under several natural models for $L^\star$ and for $X$, this quantity can easily be bounded. For example, if we treat $X$ as fixed and assume that the singular vectors of the rank-$r$ matrix $L^\star$ are chosen uniformly at random, one can show that $\|L^\star X^\top X\|_\infty \sim \sqrt{d\log(nd)/m}$.