[Reviews · NeurIPS 2015]

Submitted by Assigned_Reviewer_1

1. The paper reads very well, but readers would greatly benefit from more intuition.

No intuition is given as to why recovery in the case of compressed data is possible.

Without any technical details or intuition, the paper reads mostly as a list of technical results. 2.

Alternating minimization seems fine when alpha=infinity.

But the case of finite alpha is not described at all for experimentation (or even mentioned for the Chlorine sensor data) 3. Why does the theory hold in simulation if alpha=infinity for the instances of problem (2) that are solved? 4. Regarding the Chlorine Sensor Data, a real data example is great, but the section does not say anything about how compression fits into this example.

Is compression practical for only 4310 time points and actually used in practice for such things? Or is compression done here as an exercise on real data?

It also does not say anything about what kind of compression is done.

Few typos: 1. Eliminate last comma in abstract 2. pg 3, top line: change "propose natural" to "propose a natural" 3. pg 6, line 287: change "componend" to "component" 4. pg 7, line 368: missing period after n=d=400
Summary: The paper considers robust PCA under data compression, when

the 'ground truth' combination of a low-rank and a sparse matrix is compressed by applying some compression matrix C to obtain the observed data (for efficiency,

privacy, or other purposes). The problem is interesting and the paper presents theoretical results as well as simulations and some results on a practical problem.

Submitted by Assigned_Reviewer_2

The paper is extremely well-written, the problem at hand is clearly presented.

Specific comments:

\alpha in equation (2) needs to be more clearly defined apriori.

A few examples and domains for the different cases of the Compression matrices in Section 2.2 will be helpful.

A little more could be said about the importance of each of the theorems and the implications of the error bounds for each case scenario in section 3.2 and 3.3, for immediately transitioning to experiments.

Results are clear for figure 1 but not so for figure 2. If the linear scaling needs to be clearly scene, the variables for the x-axis and the different curves need to be swapped? i.e. it may be better to plot against the m dimension for different sparsity levels or rank. It is hard to see the linear scaling in this figure. What is the implication for deviation from linear scaling for small m?

Results from the chlorine data perhaps could be presented with other benchmarks to show performance improvements - which appear small in the current presentation.

Proofs of the theorems are very elegant.
Summary: This paper examines the robust PCA problem under data compression to determine a low-rank and sparse subspace matrix decomposition of a compressed matrix. A convex program for recovering the sparse and low-rank components are derived. Importantly upper bounds on error are derived for additive noise or missing data, which scales linearly with dimension, compression and complexity. Empirical results are presented for both simulations and a chlorine sensor dataset,

Submitted by Assigned_Reviewer_3

This paper proposes a convex robust PCA method that differs from previous work in that it is designed to be applied to compressed data (i.e. to a lower-dimensional linear transformation of the data).

The contributions include the problem statement, its formulation as a convex program, a theoretical approximation bound, specializations for specific assumptions on the noise and compression matrix, empirical results on small synthetic data illustrating the rates of convergence are in accordance with theory, and one small synthetic result providing circumstantial evidence that the approach allowing for sparse perturbations appears beneficial to standard low-rank matrix factorization on the data at hand.

The work presented includes non-trivial results that require a high degree of sophistication. The method itself is new and possibly important in a number of practical applications.

On the other hand, none of the contributions are breaking new ground. E.g. the problem statement and formulation are straightforward given the literature on compressed sensing and convex approaches multi-task learning and multi-variate regression. The theory, while by no means trivial, contains results of a similar kind and relies on existing proof techniques presented in earlier work (e.g. [1]).

The experiments raise questions regarding scalability (surely it is possible to solve this optimization problem for larger data sets?). While they do illustrate the rate of convergence is similar to what the bounds predict, it would have been useful to also see to what extent the constants in the bound are tight (e.g. by plotting the actual bounds on top of the same figures).

Summary: The paper proposes a straightforward version of robust PCA for compressed data, combined with theoretical results (approximation bounds) the proving of which requires advanced (though existing) proof techniques. Some initial empirical results are included.

Author Feedback
Author rebuttal: We would like to thank the reviewers for their helpful comments.
As briefly discussed in the introduction, we are mostly motivated by the ideas in compressed sensing and randomized numerical linear algebra literature, where the signals with low-dimensional structure (e.g. sparsity or low rank) can be perfectly or approximately recovered from random projection measurements. Since robust PCA is the problem of demixing such two superimposed structures from data, we thus should be able to apply the idea of random compression into robust PCA context, which can play a significant role in today's big data era, where data-size and privacy are of great concern in data analysis. We will emphasize this more in our next draft.

Reviewers 6 and 8 both asked about the \alpha parameter. It is related to the so called 'spikiness' condition, which has been developed in matrix completion and matrix decomposition literature (e.g. [1]) to control the interaction between low rank and sparse components. As can be seen clearly in the proof of theorems in section 3.2 and 3.3, if the underlying true low rank component is under the spikiness condition, the compressed low rank component also follows the similar spikiness condition. In practice, however, running the algorithm with \alpha=\infinity is simpler and generally results in a solution that still has low spikiness (i.e. the solution is the same as if we had imposed a bound with a finite \alpha). We will clarify this in our next draft.

To address Reviewer 8's question about Figure 2, although we have a little deviation from linear scaling for small m in figure 2, this can be due to the fact that the theorems give upper bounds rather than tight matching upper and lower bounds, or perhaps small values of m do not yet satisfy the condition stated in the theorems. However, the plots do show approximate linear scaling that can be well explained by our theory.